# Plasma membrane flipping of Syntaxin-2 regulates its inhibitory action on insulin granule exocytosis

Fei Kang [1,2] ✉, Li Xie[1], Tairan Qin[1], Yifan Miao[1], Youhou Kang[1], Toshimasa Takahashi [1], Tao Liang[1,2], Huanli Xie[1] & Herbert Y. Gaisano [1,2] ✉

Enhancing pancreatic β-cell secretion is a primary therapeutic target for type-2 diabetes (T2D). Syntaxin-2 (Stx2) has just been identified to be an inhibitory SNARE for insulin granule exocytosis, holding potential as a treatment for T2D, yet its molecular underpinnings remain unclear. We show that excessive Stx2 recruitment to raft-like granule docking sites at higher binding affinity than pro-fusion syntaxin-1A effectively competes for and inhibits fusogenic SNARE machineries. Depletion of Stx2 in human β-cells improves insulin secretion by enhancing *trans*-SNARE complex assembly and *cis*-SNARE disassembly. Using a genetically-encoded reporter, glucose stimulation is shown to induce Stx2 flipping across the plasma membrane, which relieves its suppression of cytoplasmic fusogenic SNARE complexes to promote insulin secretion. Targeting the flipping efficiency of Stx2 profoundly modulates secretion, which could restore the impaired insulin secretion in diabetes. Here, we show that Stx2 acts to assist this precise tuning of insulin secretion in β-cells, including in diabetes.

Insulin secretory deficiency in Type-2 diabetes (T2D) is contributed by defective secretory granule (SG) exocytosis attributed to deficiency of soluble N-ethylmaleimide–sensitive factor attachment protein receptor (SNARE) proteins[1]. The SNARE machinery consists of 3 families, vesicle-associated membrane proteins (VAMPs), synaptosomal-associated protein 23/25-kDa and syntaxins (Stxs), which come together in a *trans*-SNARE complex that docks SGs onto the plasma membrane (PM) in a hemifused state[2,3]. After stimulated SG fusion with PM, the SNARE complex assumes a *cis* configuration that undergoes disassembly to recycle component SNAREs for another round of SG fusion[2]. The cycle of SNARE complex assembly and disassembly is tightly orchestrated by many regulators and chaperones[4]. Glucose-stimulated insulin secretion (GSIS) in pancreatic β-cells is characteristically biphasic, with initial peak phase lasting 15 min contributed largely by fusion of docked SGs, and the subsequent phase lasting hours arise from newcomer SGs undergoing minimal residence time at the PM before fusion[4,5]. How insulin SGs are fated to be docked vs. newcomer is dictated by appropriate pairing of syntaxins (Stx1a for docked, Stx3 for newcomers) and VAMPs (VAMP2 for docked, VAMP8 for newcomers)[5–8].

Not all SNAREs promote SG fusion. SNAP23, the non-neuronal isoform of SNAP25 that mediates exocytosis in non-neuronal cells, paradoxically acts as an inhibitory SNARE (i-SNARE) in β-cells[9,10]. Stx2, also an i-SNARE, its deletion promoted insulin SG[11] and pancreatic acinar exocytosis[12]. However, the precise mechanism by which Stx2 acts as an i-SNARE remains undefined. The 34-kDa Stx2 was better studied as an extracellular morphogen epimorphin involved in epithelial morphogenesis in skin and lung[13,14]. Stx2 exists in three configurations, a full-length form located in both inner and outer PM surface and an extracellularly-truncated form of 30-kDa[15]. This is brought about when intracellular Stx2 flips across the PM, followed by extracellular cleavage at H246 to release the 30-kDa epimorphin[14,15]. Stx2/epimorphin's morphogenic property is mediated by its Habc domain[16], not the H3/SNARE domain that interacts with cognate SNARE proteins to mediate fusion[2,3].

In this work, we assessed the significance of this PM flipping of Stx2 in regulating its i-SNARE action on SG exocytosis. Here, we show that modulating PM flipping of Stx2 from inside (inhibitory) to the exterior (relief of inhibition) of the β-cell assists

[1]Department of Medicine, Temerty Faculty of Medicine, University of Toronto, Toronto, ON M5S 1A8, Canada. [2]Toronto General Hospital Research Institute, University Health Network, 200 Elizabeth Street, Toronto, ON M5G 2C4, Canada. ✉e-mail: fei.kang@utoronto.ca; herbert.gaisano@utoronto.ca

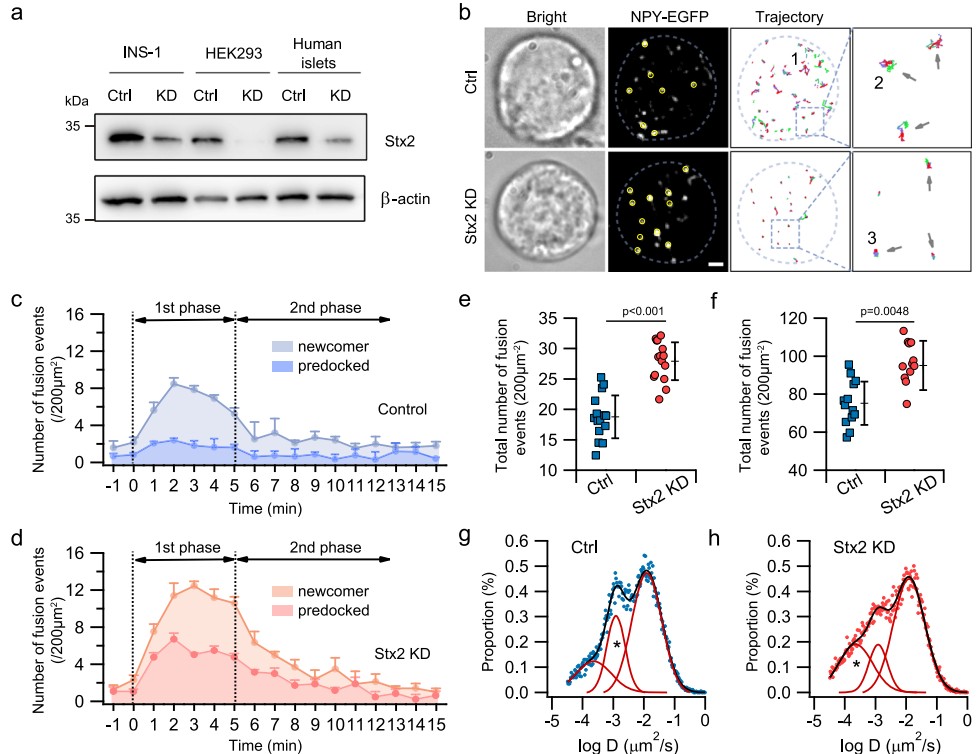

**Fig. 1 | Stx2 deletion increases exocytosis of predocked and newcomer granules in human β-cells. a** Representative western blot ($n = 3$ independent experiments, analysis in Supplementary Fig. 1a) showing the efficiency of Stx2 KD (see Methods) in human cells (HEK293A cell, human islets) and rat β-cells (INS832/13). β-actin used as loading control. **b** Representative TIRFM images monitoring the behavior of granules with the fluorescent granule marker neuropeptide-Y-EGFP (NPY-EGFP). Yellow circles indicate the sites of granule fusion events for Stx2-KD and Control (Ctrl) human β-cells. Granule trajectories are shown on the *right* with colors code of the times elapsed. These images are representative of $n = 10$ cells for Control and Stx2-KD from three independent experiments, from which the number of fusion events (**c–f**) and analysis of granule mobility (**g, h**) were assessed. Scale bar, 1 μm. The number of fusion events of predocked and newcomer in control (**c**) and Stx2 KD (**d**) human β-cells at 1-min intervals during first and second phases of glucose (16.8 mM) stimulated insulin secretion. Data obtained from 5 pancreatic islets donors and shown mean + s.e.m. Summary of fusion events of predocked

(**e**) and newcomer (**f**) granules calculated from (**c, d**), showing Stx2-KD increased the exocytosis of predocked and newcomer granules. Data shown mean ± s.d., two-tailed unpaired Student's *t*-test. Histogram of the granule diffusion coefficient D measured from the trajectory of individual granule in WT (**g**) and Stx2 knock-down (**h**) human β cell after glucose stimulation. Diffusion coefficient *D* is a measure of how constrained diffusion of the particles, smaller values indicate more restricted movement[52]. Dots represent raw data and the histogram was fitted with a three-component Gaussian function (defining 3 subpopulations indicated as 1, 2 and 3 in **b** and explained in the Results Section), sum of each peak was represented by black lines. The asterisk in **h** indicates that Stx2 KD β-cells had more granules that exhibited more restrictive movement (25.2% vs. 18.2% in Control β-cells) whereas the asterisk in **g** indicates that Control β-cells had more granules exhibiting less restrictive movement (23.9% vs. 14.1% in Stx2-KD β-cells). Source data are provided as a Source Data file.

the precise tuning of insulin secretion, and this flipping efficiency can be used to alleviate the impaired insulin secretion in diabetes.

## Results
### Syntaxin-2 deletion promotes exocytosis of docked and new-comer SGs

To determine the endogenous role of Stx2, we deleted Stx2 in human (HEK, islets) and rat β-cells (INS832/13) using specific shRNAs (in Methods) with knockdown (KD) efficiency of 85% and 90%, respectively (Fig. 1a, Supplementary Fig. 1a). Stx2-KD in islet β-cells from nondiabetic human donors (see Supplementary Table 1 for donor information) were assessed for GSIS and single-SG behavior using total internal reflection fluorescence (TIRF) microscopy (Fig. 1b-h). Compared to Control β-cells (Fig. 1c), Stx2-KD β-cells (Fig. 1d) exhibited more fusions of predocked and newcomer SGs[5] (analysis in Fig. 1e–f), in agreement with Stx2-knockout mouse β-cells[11]. To confirm, patch-clamp electrophysiology capacitance measurements (Supplementary Fig. 1b-d) showed higher exocytosis in Stx2-KD β-cells than control β-cells (Control: 63.7 ± 6.6 fF/pF; Stx2-KD: 98.3 ± 10 fF/pF).

We resolved the dynamics of SG behavior by single-particle tracking analysis[17]. Histograms of diffusion coefficients D of individual SGs (Fig. 1g–h) were used to assess how restricted SG movements are, with mean squared displacement MSD = 4DΔt estimating D of every SG in consecutive frames. With Bayesian probability theory[18], we characterized the collective distribution of diffusion coefficients D and identified three subpopulations, which fit a three-component Gaussian distribution: (a) dynamic population (D-$10^{-2}$ μm²/s; trajectory example in Fig. 1b indicated as 1) representing SGs approaching and leaving the PM; (b) confined population (D-$10^{-3}$ μm²/s; see Fig. 1b indicated as 2) corresponding to SGs bound to PM with restricted motion more characteristically seen in control β-cells; and (c) resting population (D-$10^{-4}$ μm²/s, see Fig. 1b indicated as 3) representing nearly-immobile docked SGs more characteristically seen in Stx2-KD β-cells. Comparing these histograms, Stx2-KD β-cells displayed a larger percentage of stably docked SGs than control β-cells, with distribution of the three subpopulations of SGs for control (Fig. 1g) being 57.9%, 23.9%, and 18.2%; and Stx2-KD (Fig. 1h) being 60.7%, 14.1% and 25.2%, respectively. These findings suggest that Stx2 deletion promotes insulin secretion by facilitating stable docking of SGs onto the PM, which is required before priming to become fusion-competent[19].

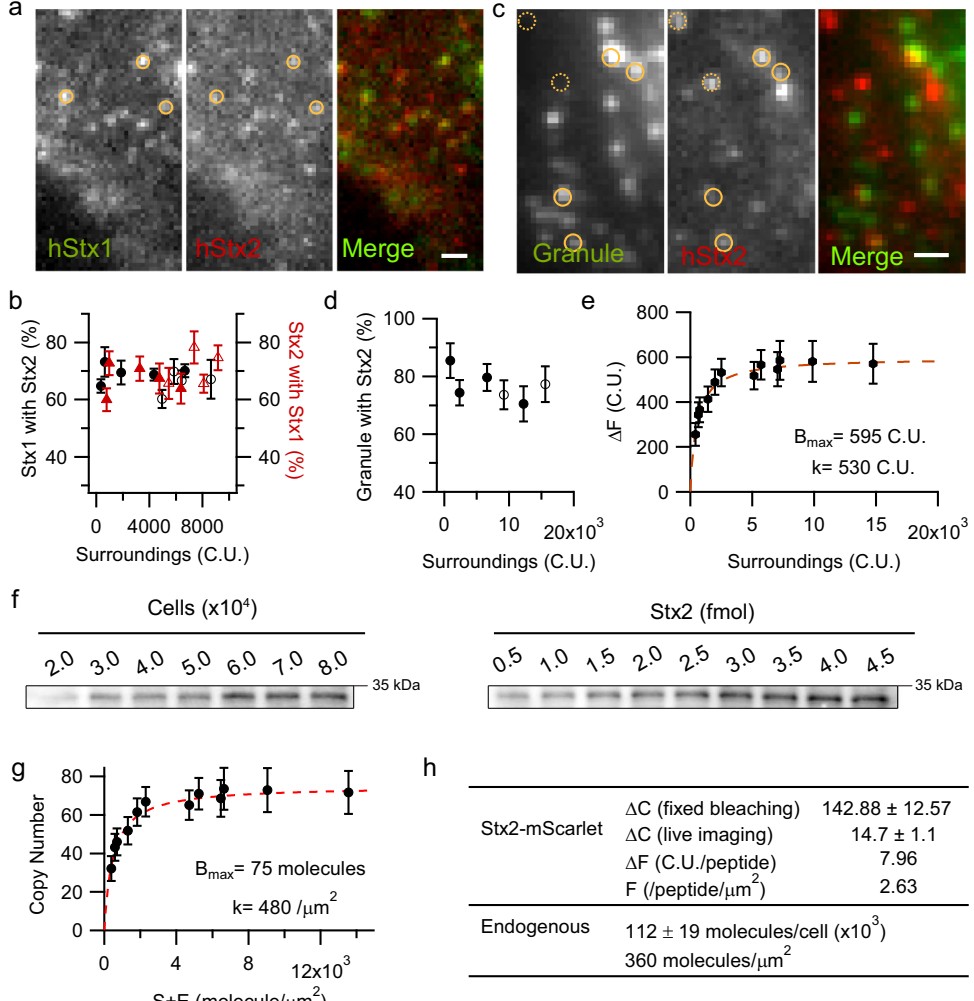

**Fig. 2 | Stx2 colocalizes and clusters with Stx1a on secretory granule docking sites. a** Representative images of β-cell sparsely coexpressing Stx1a-mNeonGreen and Stx2-mScarlet using the strategy of tandem RT-tags fused constructs. Both images averaged with 3 consecutive frames in a movie, same regions were indicated with circles ($n = 3$ independent experiments). Scale bar, 1 μm. **b** Relationships between Stx2 with Stx1a as expression levels are increased. Solid symbols indicate measurement with RTs strategy, empty symbols by CMV promoter. **c** Stx2 clusters beneath the secretory granules. Human β-cell coexpressing NPY-EGFP and RTs-Stx2-mScarlet. The circle indicates granule-associated Stx2 cluster and dashed circle indicates granule-unrelated to Stx2. Images showed an average of every 100th frame in a movie, 5 frames in total. Scale bar, 1 μm. **d** The percentage of docked granules with Stx2 cluster remain constant regardless of expression levels. **e** The fluorescent intensity of Stx2 clusters correlates with the overall expression level. The fluorescent intensity of granule-associated Stx2, $\Delta F$, varied with that of granule-unrelated Stx2, surrounding annulus, S. The curve is fitted by a reversible protein-binding to fixed target sites model, $\Delta F = B_{max} \times S/(k + S)$, as the dashed red line indicated, $B_{max} = 595$ C.U. and dissociation constant, $k = 530$ C.U. **f** Quantitative western blots determine the endogenous copy number of Stx2 by comparing purified recombinant Stx2 in human β-cell lysates. Linear dilutions of known amounts of cell lysate (*left*) and recombinant Stx2 (*right*) used as the standard in amounts ranging from 0.5–4.5 fmol were included on the same gel ($n = 3$ independent experiments). Human β-cells recognized by HPi2+/HPa3− antibody and counted by Fluorescence-Activated Cell Sorting (FACS) (see Methods). We estimated that the overall mean copy number of Stx2 is $112 \pm 19 \times 10^3$/cell, that is, ~360 molecules/μm². **g** Copy number within single Stx2 cluster saturates as the global expression levels increase. Copy number converted from intensity in **b** by the corresponding factor shown in rows 3,4 of (**h**). $B_{max} = 75$ molecules (derived from 595/7.96 = 75) and $k = 480$ /μm². When Stx2 expresses endogenously, $E = 360$ molecule/μm², the endogenous copy number of Stx2, $B_{endo} = 32$ molecules (derived from 75*360/ (480 + 360) = 32) bound per cluster. Data are expressed as mean ± s.e.m. in **b**, **d**, **e**, and **g**. **h** Summary of single-molecule properties and endogenous abundance of Stx2. $\Delta C$, the mean bleaching step size of single Stx2-mScarlet; $\Delta F$, the conversion factor for cluster intensity-to-copy number. Source data are provided as a Source Data file.

## Syntaxin-2 colocalizes with but at higher binding affinity than syntaxin-1a clusters on SG docking sites

We postulated that Stx2 effects on SG fusion might be related to potential interactions with Stx1a, with which it shares 68% identity and 82% homology[16,20]. Combining translational readthrough (RT) strategy[21] with viral gene delivery system, we introduced tandem RT-tags fused gene of interest into human β-cells to precisely control protein expression and enable in vivo long-term single-molecule imaging. We investigated the subcellular location of Stx2 in relation to SGs and Stx1a (known and confirmed here to form raft-like clusters that serve as docking receptors for SGs[19]). Stx2 (Stx2-mScarlet) was located in the PM forming punctate clusters similar to Stx1a[22], and in fact colocalized with Stx1a (Stx1a-mNeonGreen, Fig. 2a). Stx2, at low expression, began to appear as clearly visible puncta, then became increasing more saturated within each cluster as Stx2 expression was increased, indicating recruitment of Stx2 molecules into the puncta to form larger clusters. At higher expression, Stx2 clusters became oversaturated and indistinguishable from the surrounding since the raft-like docking receptor could no longer bind excess Stx2 molecules which then distributed more homogeneously on the PM. We assessed whether increasing Stx2 or Stx1a expression could increase their colocalization in PM (Fig. 2b). On average, 70−75% of Stx2 clusters were

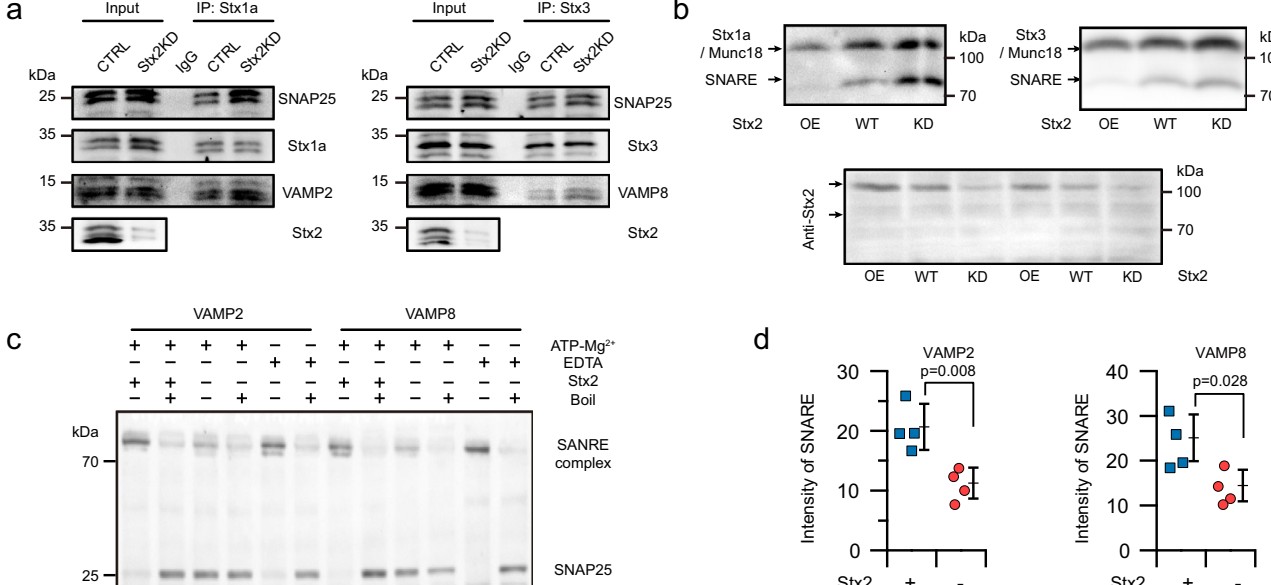

**Fig. 3 | Stx2 deletion enhances exocytosis by facilitating *trans*-SNARE complex assembly and *cis*-SNARE complex disassembly. a** Stx2 deletion promotes the formation of profusion *trans*-SNARE complexes. SNARE complexes were co-immunoprecipitated (co-IP) from human islets lysates by antibody against Stx1a and Stx3. Native primary antibody-specific secondary antibodies were used to eliminate interference by the heavy- and light-chain IgG fragments of initial IP assay. Representative blots are shown in **a** and quantification shown in Supplementary Fig. 3a (*n* = 4 independent experiments). **b** Stx2 inhibits the transition of early assembly intermediates to fusogenic SNARE complexes by substituting for the fusogenic syntaxins (Stx1a, Stx3) as detected by native-PAGE. Samples were loaded with 5× native loading buffer into 12% non-denaturing polyacrylamide gel without SDS. This is because Stx2 directly bound not only cognate SNAREs but also the

priming factors, Munc18a and Munc13 (in Supplementary Fig. 3b) (*n* = 3 independent experiments). **c, d** Stx2 deletion enhances disassembly of *cis*-SNARE complexes. Native SNARE complexes isolated from INS832/13 by antibody against VAMP2 or VAMP8 were disassembled by recombinant NSF and α-SNAP in the absence or presence of Stx2. 10 mM EDTA was used to inactivate ATP hydrolysis by chelating $Mg^{2+}$. SDS-resistant complexes disassemble completely when the sample was heated to 100 °C before electrophoresis. SNARE complexes were visualized by SDS/PAGE and immunoblotting using antibody against SNAP25. Representative blots are shown in **c** and quantification in **d**. Values are mean ± s.d. from 4 independent experiments (two-tailed unpaired Student's *t*-test). Source data are provided as a Source Data file.

Stx1a-positive and vice versa, regardless of expression level of either Stx. When Stx2 was coexpressed with SG marker NPY-EGFP (Fig. 2c), Stx2-mScarlet clusters aggregated beneath the docked SGs (closed circles). We also found some Stx2 clusters were not on SGs (dashed circles), and some SGs were not on Stx2 clusters. In fact, regardless of the Stx2 expression level, ~75% of docked SGs were associated with Stx2 (Fig. 2d).

We quantified the cluster behavior of Stx2 using M. K. Knowles, et al.'s strategy[22] (Fig. 2e). To calculate the copy number of Stx2 clusters, we quantitated the brightness of individual fluorescently-labeled Stx2 molecules and overall average endogenous Stx2 copy number. In spite of the diffraction limit of the imaging system, characterization of stepwise bleaching of single fluorophore can distinguish individual Stx2 molecule (Supplementary Fig. 2a, b). By bleaching fixed Stx2-mScarlet in β-cells (Supplementary Fig. 2a–c) along with performing quantitative Western blots (Fig. 2f), we determined the brightness of single mScarlet and abundance of endogenous Stx2 (summarized in Fig. 2h) and Stx1a for comparison (Supplementary Fig. 2d, e). On average, endogenous Stx2 expresses on human β-cell surface at a density of 360 molecules/$\mu m^2$ (Fig. 2g), which is 40% lower than Stx1a (600 molecules/$\mu m^2$, Supplementary Fig. 2d). Due to a relatively higher binding affinity, an approximately equal copy of Stx2 (32 molecules, Fig. 2g) and Stx1a (34 molecules, Supplementary Fig. 2d, e) are assembled into raft-like clusters. The copy of Stx2 within these nano-size docking clusters is ~10× more than the estimated minimum copy of SNARE complexes required to mediate single SG fusion[23]. Taken together, endogenous Stx2 is expressed in relative excess and binds raft-like docking sites with somewhat higher affinity than Stx1a.

Since SNAP23 in β-cell acts as an i-SNARE to inhibit SG exocytosis by blocking SNAP25 interaction with voltage-gated calcium channels

($Ca_Vs$)[10], we postulated that Stx2 could also alter $Ca_Vs$ directly in competition with Stx1a. However, patch-clamp electrophysiology demonstrated that Stx2-KD in human β-cells compared to control β-cells showed no differences in $Ca_V$ opening kinetics or channel density (Supplementary Fig. 2f–i). This led us to the hypothesis that Stx2 affects the SNARE fusion machinery.

## Syntaxin-2 deletion increases exocytosis by facilitating both the core *trans*-SNARE complex assembly and *cis*-SNARE disassembly

To respond to continual assimilation of nutrients after a meal that could cause deleterious sustained increases in blood glucose, hours-lasting GSIS have to be maintained, which requires repeated rounds of insulin SG fusion involving repeated cycles of SNARE protein assembly and disassembly[1,4]. We therefore explored how Stx2 may be involved not only in SNARE complex assembly but also disassembly. SNARE complexes in β-cells are Stx1a/Munc18a/SNAP25/VAMP2 for pre-docked SGs and Stx3/Munc18b/SNAP25/VAMP8 for newcomer SGs[1,5]. To determine whether Stx2 depletion affects the assembly of native SNARE complexes, we immunoprecipitated endogenous SNARE complexes from detergent-solubilized extracts and probed the precipitate for individual SNAREs, which showed an increase of 55% for VAMP2 and 62% for SNAP25 by Stx1a antibody pull-down; and an increase of 38% for VAMP8 and 45% for SNAP25 by the Stx3 antibody pull-down (Fig. 3a and Supplementary Fig. 3a). Single-molecule imaging and co-IP study of human islets (Supplementary Fig. 3b) suggested that Stx2 functions as i-SNARE by directly competing for and binding to a subunit of SNAREpins to form a non-fusogenic complex that included priming factors Munc18 and Munc13, consistent with our previous pull-down and co-IP study in mouse[11]. Munc13-catalyzed conformational changes have been shown to be essential for the transition of the

Munc18/Syntaxin complex to the full primed fusion-ready SNARE complex[11,24]. Western blots showed that Stx2 knockdown does not affect the expression of Munc13 and Munc18 (Supplementary Fig. 7d), indicating that the catalytic (priming) chaperones for SNARE complex assembly remain unchanged. In contrast, the native-PAGE study showed Stx2 inhibits the transition of early assembly intermediates to fusogenic SNARE complexes by substituting for fusogenic syntaxins (Stx1a and Stx3, Fig. 3b). This is confirmed by Stx2 pulling down not only cognate SNAREs but also the priming proteins Munc18 and Munc13 (Supplementary Fig. 3b). Next, we investigated whether Stx2 plays a role in the disassembly of cis-SNARE complexes. A complete set of cognate SNARE proteins characteristically form very stable complexes that are resistant to SDS (sodium-dodecyl-sulfate) and can tolerate temperatures of ~80 °C without significant denaturation, but lack of any component leads to loss of SDS-resistance and thermal stability[25]. These properties enabled exclusion of binary complexes and more precisely quantify core ternary SNARE complexes than conventional co-immunoprecipitation. Native SNARE complexes from insulinoma INS8332/13 lysates were isolated by incubation with anti-VAMP2 or VAMP8 antibody-coated superparamagnetic beads followed by extensive washing, then purified α-SNAP and NSF added in presence of ATP-Mg$^{2+}$ or EDTA (Fig. 3c). After Stx2-KD, more SNARE complexes disassembled (Fig. 3c, d), indicating that Stx2 functions as an inhibitory regulator of the disassembly of cis-SNARE complexes. Therefore, Stx2 deletion would facilitate the repeated rounds of cis-SNARE complex disassembly (Fig. 3c, d) required to replenish the components for assembly of trans-SNARE complexes, also favored by Stx2-KD (Fig. 3a, b); thus, together should enable the more sustained GSIS.

## Syntaxin-2 flips from the inner to outer surface of the cell membrane in human β-cells

To selectively label and visualize the inside-out flipping of Stx2 on β-cell PM, we designed a genetically encoded sensor by fusing fluorogen-activating protein (FAP)[26,27] to cytosolic N-terminus of Stx2, whose fluorescence increases dramatically only when it flips out to the extracellular surface and non-covalently binds to an extracellular PM-impermeant fluorogen (Fig. 4a). This binding is reversible when FAP-Stx2 flips back into the cell interior wherein the fluorescence dampens then disappears. This provides exquisite spatiotemporal resolution of the dynamics of Stx2 flip-out/flip-in. We validated this FAP-Stx2 imaging strategy in INS832/13 cells showing discernible signals detected only at the PM surface of cells expressing FAP-Stx2 (Fig. 4b, c), wherein 80% increase in global fluorescence increase as well as localized single flip-out events (sparks) (Fig. 4c, d) could be tracked after exposing to the glucose in the presence of 100 nM fluorogen. The flipped single-clusters indicated by localized transient fluorescence bursts generally last for seconds to tens of seconds, which then dissipate on the cell surface (Fig. 4d). In the meantime, the global intensity of the FAP-Stx2 on the cells increases gradually over time (Supplementary Fig. 4c).

We critically assessed the dynamics of Stx2 flip-out in response to glucose stimulation in human β-cells. To accurately evaluate Stx2 on the extracellular surface, flow cytometry studies were performed on FAP-Stx2-expressing β-cells in the presence of 200 nM fluorogen (Fig. 4e, f). Resting β-cells demonstrated ~0.4-fold ($F_0/F_{bg}$) increase in fluorescence intensity after fluorogen incubation, suggesting that there exists flip-out Stx2 on the extracellular surface at resting state. After glucose stimulation, surface Stx2 increased steadily from time 0 to 15 min to a final ~5-fold compared with resting state (Fig. 4f). Population analysis of FAP- accessibility using flow cytometry showed a similar result as the imaging study, which suggested that ~15% of Stx2 is present on extracellular surface at resting state and the remaining 85% gradually getting flipped over 15–20 min after glucose stimulation. To assess the Stx2 flip-out spatiotemporal relationship with insulin SG exocytosis in situ, we used the more physiological (than isolated islets) human pancreatic slices[28] (Fig. 4g). Exocytotic events were monitored

by membrane-impermeant fluid-phase tracer, sulforhodamine B (SRB) (Fig. 4h). Both fluorophores were excluded from the cell interior at resting state (Fig. 4h). With glucose stimulation, fusion of SGs indicated by fluorescent SRB hotspots on PM had a remarkable spatio-temporal correlation with the marked increases of FAP-Stx2 flip-out, shown in enlarged images (Fig. 4i and Supplementary Fig. 4f) and a graphical analysis of a representative hotspot (Fig. 4j). These results demonstrate that flipping-out of Stx2 occurs physiologically in human pancreatic β-cells and should play a role in regulating SG exocytosis, shown next.

## Flip-out relieves syntaxin-2 inhibition of the fusogenic SNARE complex which enhances insulin secretion

We assessed the importance of this flip-out property of Stx2 on exocytosis by expressing both FAP-Stx2 and SG fluorophore NPY-tdOrange2[29] (Fig. 5a) using TIRFM imaging, which showed that localized Stx2 flipping was spatio-temporally related to triggering SG exocytosis. Generally, flipping events occur more frequently and distribute more widely than fusion events. Most (~80%) fusion events occurred at sites where flipping was occurring, whereas the proportion of flipping events occurring at fusion sites was less than 25%. We assessed how Stx2 flipping affects SG fusion with a simple kinetic model that flip-out stochastically triggers SG fusion, which demonstrated the dynamics of Stx2 flipping profoundly affected SG fusion in a delay-time-dependent manner (Fig. 5a, b), whereby extension of the delay decreases fusion probability, exhibiting a ~2.5 s half-maximal delay (Fig. 5c). SG fusion probability at the sites of Stx2 flipping within 2.5 sec delay was ~2× higher compared to the longer delay in Stx2 flipping time (Fig. 5d). We proposed that Stx2 flipping-out at raft-like docking receptors increases probability of exocytosis by reducing nanodomain concentration of Stx2 and relieving its localized inhibition of the SNARE fusion complex. The low probability at longer temporal delay may be attributed to the dynamic re-assembly of Stx2 cluster at SG docking sites. We therefore investigated the dynamic of Stx2 clusters, showing Stx2 rapidly assembled and disassembled at stationary nanodomains (Fig. 5e, f) and most (85%) Stx2 molecules spend <2.5 s to re-assemble into clusters (Fig. 5g). These results, together with Fig. 3, would invoke a model in which inside-out flipping of Stx2 transiently relieves its localized inhibition on the SNARE fusion complex to trigger SG exocytotic fusion.

We interrogated whether fine-tuning the flipping efficiency of Stx2 could affect GSIS, whereby an attenuated flipping impairs secretion whereas enhanced flipping increases secretory probability; with that latter helping restore the impaired insulin secretion in T2D. In addition to localized lipid environments, transmembrane protein topology is determined primarily by hydrophobicity of the trans-membrane domain (TMD) and inside-positively charged residues of extra-membrane domain (EMD) flanking the hydrophobic TMD, summarized as "positive-inside rules"[30,31]. Both lipid environment disturbance[32,33] and charge alteration of EMD[34] possess the capability of flipping the integral membrane protein. We reasoned that artificially changing the charged residues within the EMD would fine-tune the flipping of Stx2. To enhance flipping, we balanced the inside-positively charged residues by mutating arginine or lysine in the SNARE domain closed to the TMD (Stx2-mSNARE). Conversely, introducing more positively charged residues in the linker region (Stx2-mLNK) was expected to reduce flipping. To block the flipping completely, we replaced the TMD without affecting its target to PM with terminal polybasic lysine-rich (K) domain of STIM1 reported to help STIM1 bind to PM[35] (Stx2-polyK) (Supplementary Fig. 5). Using flow cytometry, the flip-out capability of these chimeras was assessed on a large scale (Supplementary Fig. 6), which indeed demonstrated these chimeras fine-tuned Stx2 flipping (Fig. 6a), indicating that hydrophobic TMD and net charge of inside EMD are important in determining the flipping capability of Stx2.

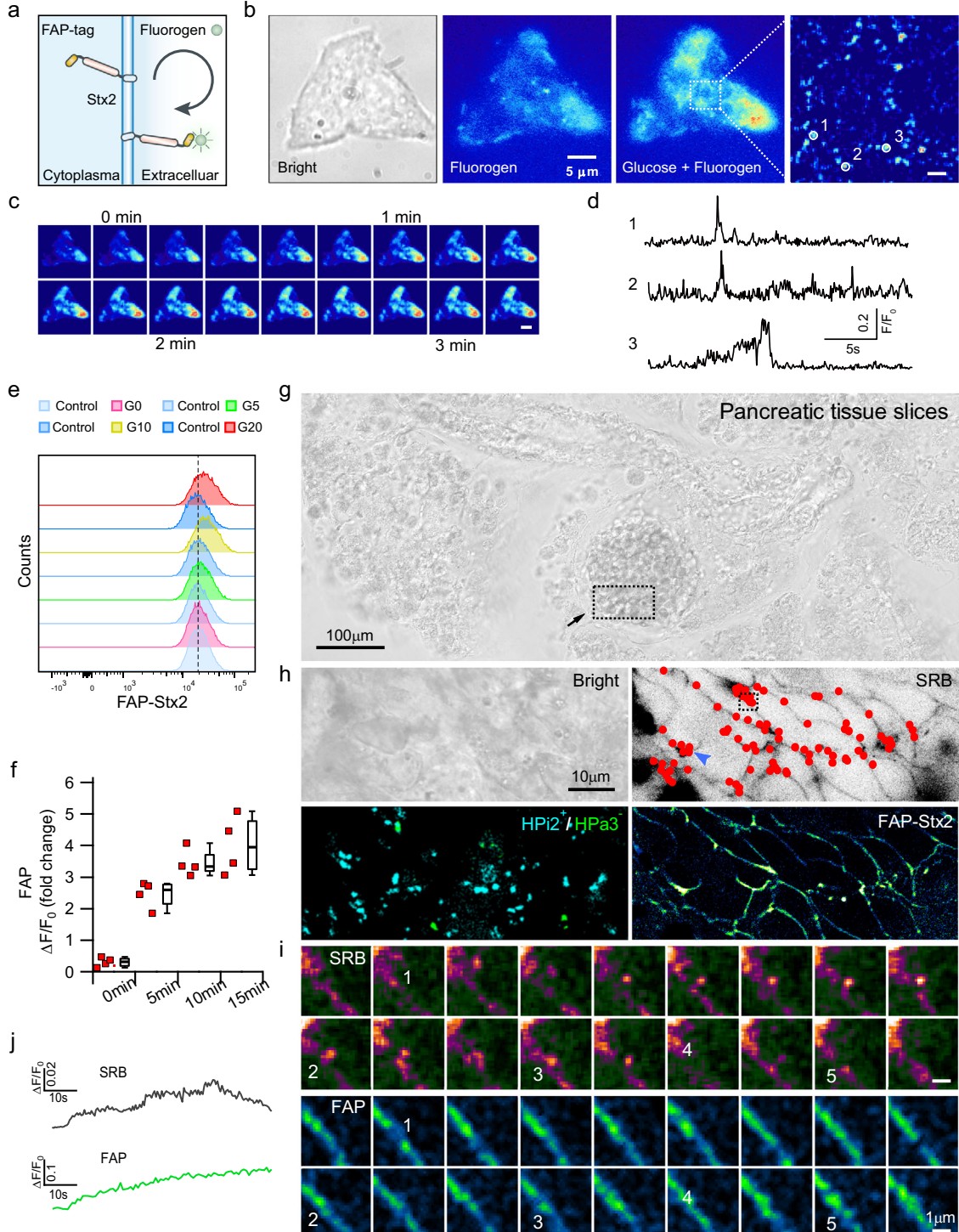

**Fig. 4 | Stx2 flips from the inner to the outer surface of the cell membrane in human β-cells. a** Schematic representation of the genetically encoded flip reporter for monitoring Stx2 flipping. **b** INS832/13 cells were infected with FAP-Stx2, stimulated with high glucose and visualized by TIRF microscopy. **c** Time-lapse images show the global Stx2 flip-out of the cell in **b**. Scale bar, 5 μm. Montages are shown at ~10-s intervals. **d** Representative time course of single Stx2 flipping events on INS832/13 cell surface ($n = 3$ independent experiments). **e** Assessment of Stx2 on the extracellular surface of single human β-cells analyzed by flow cytometry. **f** Quantification of Stx2 flip-out with the results from **e** ($n = 4$ independent experiments, box plots indicate median, 25th, 75th percentile and whiskers min/max limits). **g** Light micrograph of fresh human pancreatic tissue slice. **h** Simultaneous imaging of Stx2 flip-out (FAP) and exocytosis (aqueous tracer SRB) in human pancreatic tissue slices. Human pancreatic tissue slices infected with FAP-Stx2 for 48 h, then labeled with primary and secondary antibodies (see methods). ROI

(dashed box indicated in **g**) was selected to speed up image acquisition, preferably at β cell-rich region (*top left*). A superimposed image of SRB fluorescence shows the distribution of exocytotic events in an intact islet stimulated with 16.7 mM glucose. The dots indicate sites at which exocytotic events occurred (*top right*), which coincided with the FAP-Stx2 hotspots. Pancreatic β cells (identified by HPi2+/HPa3−) were recognized by co-labeling with antibodies against HIC1-2B4 and HIC3-2D12 (*bottom left*) ($n = 3$ independent experiments). **i** Time-lapse images of the region indicated by the dashed box in (**h**, *top right*) shown that glucose-induced exocytosis visualized by SRB fluorescent hotspots occurred where the FAP signals accumulated. Montages are shown at 3.2-s intervals. The numbers indicate the multiple exocytotic events that matched the FAP hotspots. **j** Time course of the concurrent increase in the fluorescence of SRB and FAP-Stx2 of one of these matched hotspots, indicated by the blue arrowhead in (**h**, *top right*).

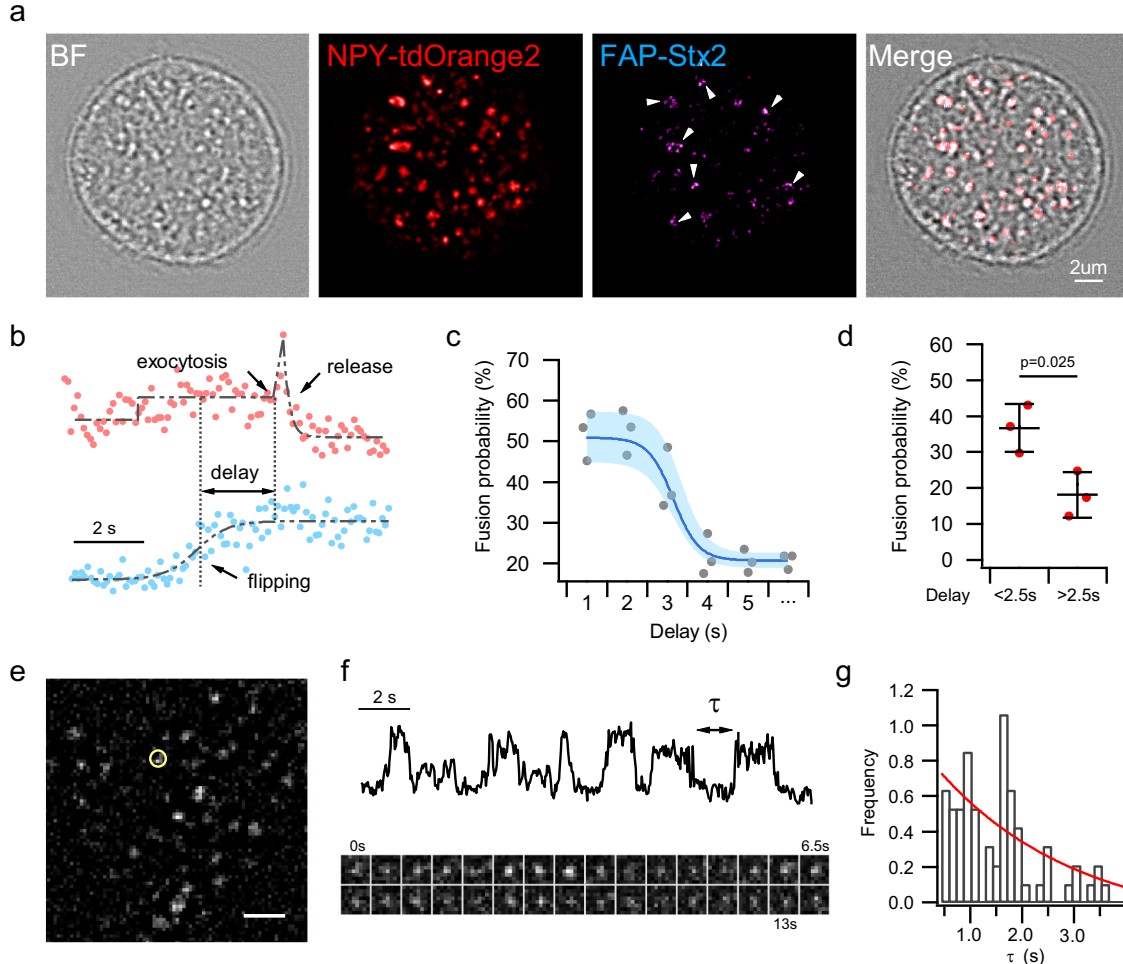

**Fig. 5 | Flip-out relieves Stx2 inhibition of the fusogenic SNARE complex.**
**a** Representative images of human β-cell infected with FAP-Stx2 and NPY-tdOrange2. White arrowheads indicate sites where exocytosis and flipping occurs (*n* = 3 independent experiments). **b** Representative fluorescence time course of single FAP-Stx2 flipping events followed by exocytosis, the latter characterized by a transient peak increase and rapid decline. **c** Relationship between delay time and exocytosis probability at fusion sites. Time delay defined by rising phase of Stx2 flipping and onset of NPY-tdOrange2 as shown in **b**. Symbols denote average fusion probability measured by the proportion of predocked and newcomer granules undergoing fusion at sites where flipping occurring; solid line shows the fitting. The shaded areas indicate ± s.e.m. The bottom axis "…" denotes delay time longer than

5 s. **d** Average granule fusion probability at sites of Stx2 flipping. Exocytosis occurred immediately after flipping within a delay time of 2.5 s vs. that of longer delay time. Each dot denotes an average probability from three independent experiments, data are presented as mean ± s.d., two-tailed unpaired Student's *t*-test. **e** Representative live-cell single-molecule imaging of Stx2-mScarlet showing that Stx2 molecules reversibly assemble and disassemble at nanodomain, an example depicted by the circled Stx2 molecule in **e** that is displayed in **f** as intensity trace and corresponding montage images. Scale bar in **e** is 2 μm. **g** Histogram of time τ that Stx2 spent to re-assemble. Source data are provided as a Source Data file.

We assessed the role of Stx2 flipping in GSIS in human islets. While insulin content remained unaffected (Fig. 6b), there were remarkable differences in GSIS (Fig. 6c, d), with FAP-Stx2-mSNARE-expressing islets exhibiting larger GSIS than WT-Stx2, and which was four-fold that of non-flipping FAP-Stx2-polyK; the later showing no increase in GSIS above the already lower basal secretion (Fig. 6c). We next assessed how these Stx2 chimeras perturb SNARE complex formation (Fig. 6e, analysis in Supplementary Fig. 7a, b). From FAP-Stx2-polyK-expressing islets, Stx1a antibody pulled down far less VAMP2 and SNAP25, and Stx3 antibody pulled down far less VAMP8 and SNAP25, when compared to FAP-Stx2 or FAP-Stx2-mSNARE. Stx1a and Stx3 antibodies pulled down more SNAP25 with FAP-Stx2-mSNARE than FAP-Stx2. Stx2 flip-out is therefore required for SNARE complexes to complete formation and effect fusion[36], and the more Stx2 flip-outs (Stx2-mSNARE > Stx2-WT > Stx2-mLNK), the more fusion occurs and presumably and consequently more frequent cycles of SG fusion would also occur; and that inhibiting Stx2 flip-out entirely (Stx2-polyK) would severely reduce secretion.

We devised a strategy wherein Stx2 that is flipped out would not be able to flip back in, which should predictably increase GSIS for Stx2 chimeras that can flip out but not for non-flipping Stx2-polyK. We took advantage of the property of botulinum toxin-C1 light chain (BoNT/C1-LC), which can proteolytically cleave at a Stx2 region that has the same conserved site as in Stx1a (Lys-Ala) at the C-terminal just off the TMD[37,38], thus deleting the entire SNARE motif and Habc domains. Due to lack of the heavy chain required to transport the toxin into the cell interior[39], the recombinant BoNT/C1-LC would only cleave extracellular but not intracellular Stx2. Thus, mSNARE, mLink and WT Stx2s were cleaved by BoNT/C1-LC, which disabled their Stx2-SNARE motifs from re-entering the cell interior to block Stx1a and Stx3 SNARE complex formation, resulting in further amplification of 16 mM GSIS (Fig. 6f) by 2-2.6-fold (Fig. 6g). Non-flipping Stx2-polyK, not cleaved by extracellular BoNT/C1-LC, would have its SNARE motif retained in cytoplasm to block exocytotic SNARE complex formation, resulting in persistent inhibition of GSIS (Fig. 6f-g). Flip-out could therefore relieve Stx2 inhibition of fusogenic SNARE complexes which enhances GSIS.

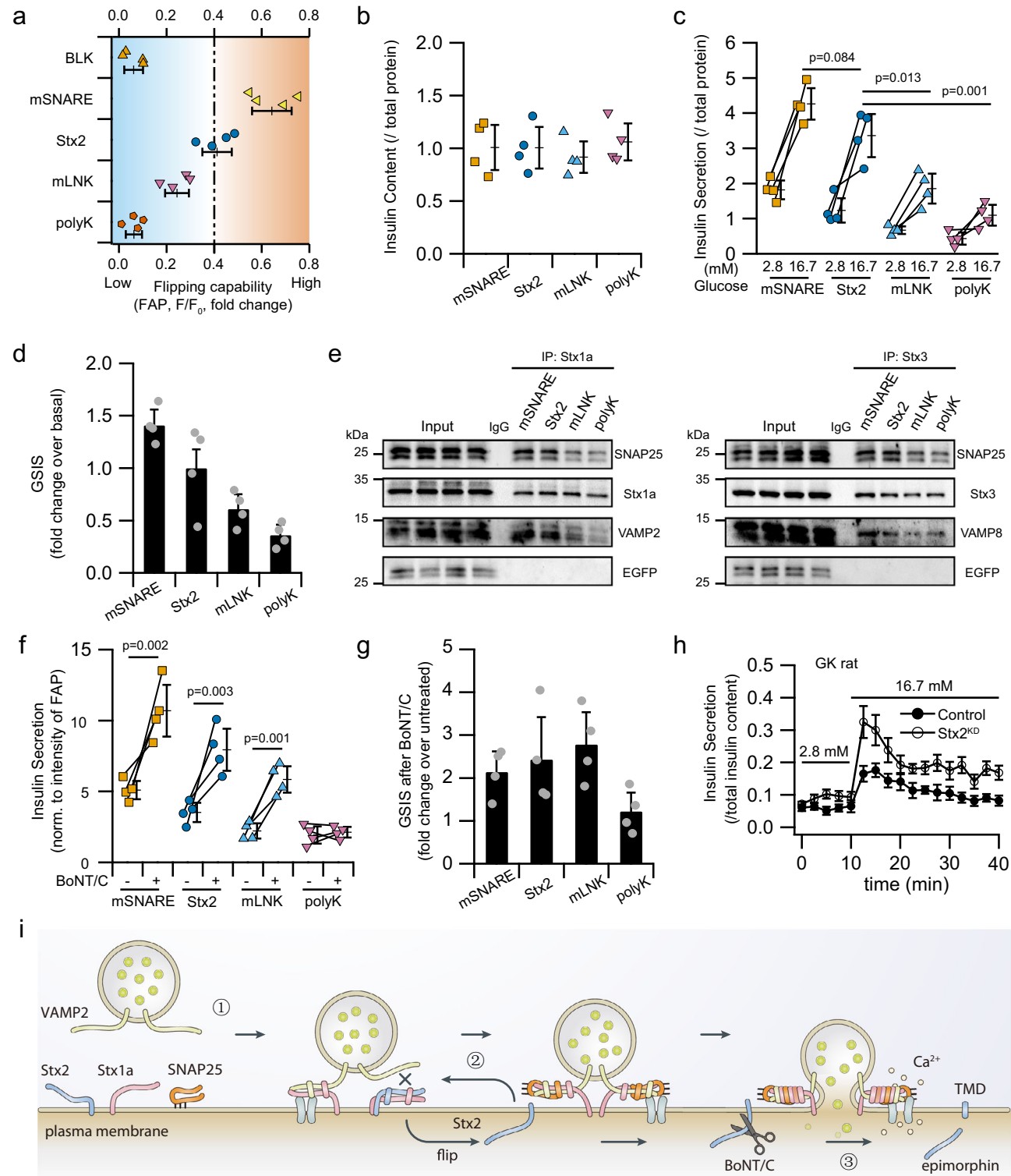

## Syntaxin-2 depletion can restore the deficient GSIS in SNARE-deficient T2D islets

Goto-Kakizaki (GK) rats very closely mimic human T2D with remarkably similar reduction in islet SNARE protein levels that in part explains the severely deficient biphasic GSIS[40,41]. Whereas levels of pro-exocytotic SNARE proteins were severely reduced in T2D human and GK rat islets[40–42], their Stx2 levels were normal[11,42], suggesting that Stx2 in T2D β-cells could be asserting an even stronger inhibition than in normal β-cells. We treated GK rat islets

with Stx2-shRNA virus (which reduced Stx2 expression in WT rat islets by 80%, Supplementary Fig. 7e), which rescued both first and second phase GSIS (Fig. 6h, AUP analysis in Supplementary Fig. 7f). Thus, in spite of greatly reduced levels of profusion SNARE proteins in T2D β-cells[40,41], Stx2 depletion must still be able to facilitate sufficient SNARE complex assembly and disassembly to replenish enough component SNAREs for repeated rounds of SNARE complex formation, to enable the rescue and sustenance of biphasic GSIS in diabetic islets.

**Fig. 6 | Modulating Stx2 flipping efficiency regulates insulin secretion.**
**a** Quantification of the flipping capability of the Stx2 mutants in INS832/13 cells by
flow cytometry. Note that mSNARE increased Stx2 flipping whereas other mutants
reduced Stx2 flipping, with the least flipping for the polyK mutant. **b** Insulin content
(normalized to normal Stx2) remained unchanged in human islets after two-round
coinfection with the indicated Stx2 mutants and Ad-shRNA-Stx2. **c** Infection with
mutants targeted to affect Stx2 flipping affected glucose-stimulated insulin secre-
tion. Human islets were treated for 1 h with basal (2.8 mM) glucose versus high
(16.7 mM) glucose stimulation. Values shown were normalized to the Stx2 basal
secretion. **d** Fold-change representation of stimulated over basal secretion in
**c**, shown as highest (mSNARE with the highest flipping) to lowest secretion cap-
ability (polyK with the lowest flipping). **e** Enhanced Stx2 flipping mSNARE increased
the formation of SNARE complexes for predocked (Stx1a immunoprecipitated (IP))
and newcomer (Stx3 IP) granules, whereas reduced Stx2 flipping polyK reduced
these SNARE complexes. This is representative of four experiments, with the ana-
lysis shown in Supplementary Fig. 7a, b. **f** Flipping per se of the Stx2 mutants affects

GSIS is shown by BoNT/C1-LC cleavage of extracellular Stx2 at near the C-terminal
transmembrane domain. To negate or control for potential different expression
levels, Insulin secretion was normalized to the total FAP levels after Triton X-100
permeabilization and then subject to 100 nM β GREEN-np. **g** Fold-change repre-
sentation of 16.7 mM GSIS of BoNT/C1-LC-treated islets over the corresponding
control untreated islets of each Stx2 mutants that were performed in **f**, which
showed BoNT/C1-LC-induced enhancement of GSIS of the Stx2 mutants that were
flipped efficiently vs. no enhancement in non-flipping polyK. Data shown as
mean ± s.d. in **a** to **g** from four independent experiments. **h** T2D GK rat islets, known
to have reduced levels of SNARE proteins[11], were infected with Ad-Stx2-shRNA
(knockdown efficiency shown in Supplementary Fig. 7e), which enhanced biphasic
GSIS. The data shown are mean ± s.e.m. from four independent experiments, with
area-under-the-curve (AUC) analysis of first and second-phase secretion shown in
Supplementary Fig. 7f. **i** Summary model of the i-SNARE role of Stx2 in insulin SG
exocytosis. Source data are provided as a Source Data file.

## Discussion

i-SNARE inhibits fusion by substituting for or binding to a fusogenic
SNAREpin subunit to form a non-fusogenic complex[43]. This i-SNARE
concept was first described to provide an extra layer of regulation for
countercurrent fusion occurring in the Golgi[43], and mimicked by
SNARE-like motifs of some bacterial proteins[44]. i-SNARE functions,
specifically of SNAP23[10] and Stx2[11,12], were only recently demonstrated
in vivo to regulate pancreatic islet and exocrine function, impacting
GSIS in T2D and pathologic exocytosis in pancreatitis. In β-cell, SNAP23
competed with SNAP25 for the same SNARE complexes, with only
SNAP25-SNARE complex able to optimally bind Ca$_v$s, priming proteins
(RIM2, Munc13-1), and calcium sensors (synaptotagmins) to form
optimal fusion-competent complexes[10]. However, SNAP23 is not an
i-SNARE in pancreatic acinar cells, but binds distinct SNARE complexes
to promote exocytosis and autophagosome-lysosome fusion[12]. Stx2
acts as i-SNARE in acinar cell by blocking exocytotic SNARE complex
formation, but also binds ATG16-L1 to block ATG16-L1 binding to cla-
thrin required to recruit PM for pre-autophagosome formation during
autophagy induction[12]. Stx2's i-SNARE action in β-cell, although acts
similarly in blocking SNARE fusion complexes, we here elucidated the
precise and distinct mechanism which is that Stx2 is configured as a
glucose-regulated reversible switch with its ability to flip-in and out of
the cell (model in Fig. 6i). The SNARE and TM domain of Stx2 are
required for flipping and the conserved key amino acid residue for
soluble epimorphin releasing resides in H246[15], which is not shared by
the other fusogenic syntaxins (Stx1a, Stx4) (Supplementary Fig. 4e).
Our flow cytometry assay emphasized the importance of this TMD by
showing that the charge alteration of the extra-membrane domain
affects the flipping efficiency. Furthermore, we showed that targeting
the flipping efficiency of Stx2 profoundly modulated insulin secretion,
which would benefit restoring the impaired insulin secretion in dia-
betes. i-SNAREs therefore mediate their inhibitory actions by similar
and distinct mechanisms which are cell-context specific.

*Cis*-SNARE formed by parallel posited α-helices derived from
the cognate v- and t-SNAREs within the same membrane after
fusion is disassembled through a combined effect of α-SNAP and
the ATPase NSF, which allow the SNARE proteins to be recycled[45].
The N-terminal three-helix-bundle of Stx1a has been shown
involved in interaction with accessory proteins, such as SM pro-
teins, synaptotagmins as well as complexins, during the multistep
SNARE cycling in membrane fusion[4,45]. Our single-molecule ima-
ging demonstrated that Stx2 competes with Stx1a to bind docking
sites at a higher binding affinity, combining the fact that the
sequence of Stx2 share 70% identity with Stx1a within N-terminal
domain, we interpret our results to suggest that the competitive
binding of Stx2 to α-SNAP makes the latter binding to *cis*-SNARE
less effective, thus reducing the ability of ATPase NSF to effect *cis*-
SNARE complex disassembly.

Our results present intriguing possibilities for potential therapies
for diabetes. BoNT/C1-LC cleavage of extracellular Stx2 promoting
GSIS raises the possibility that a potential drug compound that could
bind extracellular Stx2 in β-cell could increase GSIS. Caution should
however be taken with this approach as extracellular Stx2-blocking
compounds might block the cleavage site that releases epimorphin,
which could have inadvertent effects of blocking its morphogenic
action[14]; however, the latter might have desirable oncologic ther-
apeutic potential[46,47].

## Methods

### Reagents
Cell membrane impermeable β GREEN-np (Spectragenetics) recon-
stituted in HBSS, the working concentration is 100 nM for imaging,
200 nM for FACS. Botulinum Neurotoxin Type C1 was purchased from
List Biological Laboratories. Adenosine 5′-triphosphate magnesium
salt (A9187), EDTA and D-(+)-Glucose (G7021) were obtained
from Sigma.

### Plasmids, viruses, and proteins
Genes of interest were introduced into a 3rd generation lentiviral
vector, pLJM1-EGFP, which contains a puromycin resistance gene dri-
ven by a hPKG promoter. pLJM1-EGFP was a gift from David Sabatini
(Addgene plasmid # 19319; http://n2t.net/addgene:19319; RRID:
Addgene_19319)[48]. Human shRNA of Stx2 targets to 3′ UTR, 5′-GCAT-
GAAGTTTAATTAGGA-3′ and that for rat targets to 5′-GTCATCAT-
CACGGTGGAGA-3′. SiRNA flanked by sequence of 5′-AAAA and 3′-
AAAG were cloned into linearized the 3rd generation lentiviral vectors
piLenti-SiGFP (Abmgood, Canada) by BbsI. We included both negative
and untreated control for shRNA-Stx2 knockdown, of which the
negative control was designed to have no known target to rule out non-
specific effects, while the untreated control was designed using the
blank vector that both negative control and shRNA-Stx2 were cloned
into to determine the level of cell viability. Both controls were included
when designing and validating, while, unless specified otherwise, non-
targeting negative control (referred as "control") was always included
in all experiment. Recombinant adenoviruses were generated by
homologous recombination of the shuttle vector containing gene of
interest and the adenoviral backbone constructs in HEK293T packa-
ging cell line. Gene of interest seamlessly assembled into Ad-shuttle
vector by NEBbuilder. $10^6$–$10^7$ adenovirus-containing cells were see-
ded in 150 mm cell culture dish. Viruses were harvested after checking
cytopathic effects and immediately purified as well as titered. The viral
supernatant was stored at −80 °C.

PCR amplified mNeonGreen assembled into pLJM1-EGFP to gen-
erate pLJM1-mNeonGreen. To make readthrough constructs for single-
molecule imaging, we used tandem RT tags, RT1: CAATAGCAATTA and
RT3: CAATAGGGCTTA. Stx1a-mNeonGreen was obtained by inserting

PCR amplified tandem RT tags (5′-CAATAGCAACAATAGGGC-3′) and hSyntaxin 1A into pLJM1-mNeonGreen, using multiple cloning sites (MCS) of NheI and AgeI. The fragment of mScarlet was cloned into Stx1a-EGFP with AgeI and EcoRI to generate Stx1a-mScarlet. hSyntaxin 2 flanked by RT tags and NheI (5′)/AgeI (3′) cleavage sites was inserted into pLJM1-mNeonGreen to create RTs-Stx2-mNeonGreen. PCR amplified FAPβ1 and Stx2 seamlessly assembled into linearized pLJM1-EGFP by NheI/EcoRI digesting. The pLJM1-FAP-Stx2 was used as the template to generate other FAP chimera. Mutagenesis was performed using Quik-Change mutagenesis kit (Stratagene) according the manufacturer's protocols. All constructs used in this study were prepared according to standard molecular cloning approaches and verified by sequencing. Constructs containing gene of interest were transformed into E. coli DH5α (C2987H, New England BioLabs) and induced by 0.5 mM IPTG (isopropyl β-d-1-thiogalactopyranoside). Glutathione bead was used to purify the proteins following the manufacturer's instructions.

### Preparation of pancreatic islets, tissue slices, and culture

Human pancreatic islets were purchased from the IsletCore, University of Alberta, which were from institutional review board-approved donors with preoperative written informed consent for research by donors themselves and their family; the basic information is listed in Supplementary Table 1. Human islets were dispersed into single cells for TIRFM imaging and flow cytometry analysis; this is by digestion in cell dissociation buffer at 37 °C with shaking dispersal every 1–2 min[49]. Dispersed cells as well as islets were cultured in DMEM (low-glucose, Gibco) with L-glutamine, supplemented with 10% fetal bovine serum (FBS), 110 mg/L sodium pyruvate and 100 U/mL penicillin/streptomycin at 37 °C and 5% $CO_2$. Additionally, to prevent clogging, dispersed cells were filtered by nylon mesh with a pore size of 40 μm (BD Falcon) prior to loading into flow cytometry instrument.

For human pancreatic slice preparation (Information on human pancreas donors listed in Supplementary Table 2), the pancreas samples are trimmed into $3 \times 3 \times 3$ mm cubes and embedded into 3.8%, 37 °C pre-heated low-melting agarose gel immediately after retrieving. When cooling down by ice, samples were loaded onto microtome (VT1200S, Leica) and immersed by ECS buffer. 100–140-μm-thick slices were made and carefully transferred into culture media immediately for use (see refs. 28,50 for further details).

The GK rat, type-2 diabetes model, original obtained from Karolinska Institute, Stockholm, Sweden. Diabetes was confirmed at 8-10 weeks of age in the GK rats. Age-matched male Wistar (used as control, Charles River) and GK rats were studied, both of which were housed two per cage at constant room temperature with an artificial 12-hour light/dark cycle and fed standard rat chow and water *ad libitum*. All animal procedures and use of human pancreas were carried out in accordance with ethical guidelines of the University of Toronto's Animal Care Committee and Research Ethics Board of the University Health Network, Toronto, ON, Canada, and with approved IRBs.

INS832/13 (originally from C. Newgard, Duke University, Durham, NC) cells were maintained in RPMI1640 (Gibco) containing 10 mM glucose and supplemented with 10% FBS, 1 mM sodium pyruvate, 50 μM 2-Mercaptoethanol, 100 U/mL streptomycin/penicillin at 37 °C and 5% $CO_2$. In order to minimize variation associated with repeated transient transfection and increase reproducibility, all gene-transduced INS-1 cell used in this study, unless specified otherwise, were stable cell lines generated by Lentivirus antibiotic selection. Specifically, 1 mL lentivirus supernatant infected INS-1 cell in 100 mm cell culture dish when its confluence reaches ~70% in the presence of 10 μg/mL of polybrene (Millipore-Sigma). The medium was then discarded, and 10 mL of fresh medium added, incubating the cells for 6 h at 37 °C. Fresh medium containing 2 μg/ml puromycin was applied to select cells expressing the protein of interest 2 days after infection.

Single clones were picked after a 4-day 2 μg/ml puromycin selection and then maintained in medium with 1 μg/ml puromycin. Clones were validated by western blot.

### Immunofluorescence labeling

To label live human pancreatic β-cells for fluorescence-activated cell sorting (FACS) and imaging, slices or dissociated cells incubated in DMEM + 2% FBS containing antibodies (Sigma-Aldrich, 1:100 dilution) against pan-endocrine marker HPi2 (HIC1-2B4) and non-β endocrine cell marker HPa3 (HIC3-2D12) for 30 min at 4 °C. After washing with cold DMEM, cells or slices were transferred to DMEM + 2% FBS containing pre-adsorbed Alexa Fluor conjugated secondary antibody (Abcam, 1:200) for 30 min at 4 °C. To minimize background staining, 5% goat serum were used for blocking the secondary antibody. Primary antibody against ANTI-HPI2, CLONE HIC1-2B4 (Sigma-Aldrich, MABS1999-100UG, 1:500) and ANTI-HPA3, CLONE HIC3-2D12 (Sigma-Aldrich, MABS1998, 1:500) were obtained from Millipore Sigma. Secondary antibody, Goat Anti-Mouse IgM mu chain (DyLight® 550) (Abcam, ab98675, 1:1500) and Goat Anti-Mouse IgG2b heavy chain (PE/Cy7®) (ab130790, 1:1500) were purchased from Abcam.

### Flow cytometry and FACS

Cells were analyzed and sorted with BD Fortessa X20 (Becton-Dickenson). Propidium iodide (ThermoFisher, P1304MP) staining was used to label dead cells for exclusion, and cell doublets were excluded by pulse width measurement of forward scatter (FSC). Rainbow Fluorescent Particles (RFP-30-5A) were used to standardize FACS. Flow cytometry data collected with BD FACSDiva™ and the analysis was performed using the flow cytometer software FlowJo (FlowJo, LLC, Ashland, OR).

### Immunoprecipitation and Western blotting

Islets and cells lysed with RIPA Buffer (10×, Cell Signaling, #9806S) supplemented with protease inhibitor cocktail (100×, Cell Signaling, #5872). For immunoprecipitation, antibodies were coupled to Dynabeads Protein A (Life Technologies 10001D) and incubated with islets/INS-832/13 cell lysates in a total volume of 1 mL for overnight at 4 °C. The precipitated products were boiled in Laemmli sample buffer (4×, BioRad) for 10 min and analyzed by SDS-PAGE and immunoblotting. For western blotting, the precipitated products or islets/cells lysates were separated in 12–16% gradient SDS-PAGE and then transferred to nitrocellulose membranes. Membranes were blocked with 5% nonfat milk in PBS-0.1% Tween® 20 for 1 h. The separated proteins detected by primary antibodies (SNAP25, SySy, #111 002, 1:1000; Stx1a, Sigma, # SAB5500181, 1:1000, SySy, #110 302, 1:1000; VAMP2, R&D, #AF5136, 1:500; Stx2, Abcam, #ab12369, 1:500; SySy, #110 123, 1:500; Stx3, SySy, #110 033, 1:1000; VAMP8, SySy, #104 303, 1:500; β-actin, Abcam, #ab8227, 1:10,000) and horseradish peroxidase conjugated secondary antibody (Jackson, AffiniPure Goat Anti-Rabbit IgG (H + L), #111-005-144; AffiniPure Goat Anti-Mouse IgG (H + L), #115-005-003). The secondary antibodies were incubated for 1 h at a concentration of 1:2500. For IP experiments, native IgG-specified secondary antibodies (Abcam) were used to avoid the detection of the denatured primary antibody heavy and light chains at 1:1000 dilution for 1 h at room temperature. All blots were imaged by ChemiDoc XRS + System (Bio-Rad). Fiji (https://fiji.sc/) was used to quantify the band intensity and do densitometric analysis.

### Glucose-stimulated insulin secretion

After two-round infection with virus, human islets were washed extensively with Dulbecco's phosphate buffered saline (DPBS) and then equilibrated for 30 min at 37 °C incubator with Krebs Ringer Buffer (KRB) + 2.8 mM glucose. Cells were stimulated with 16.7 mM glucose for 1.5 h at 37 °C. Supernatant/cells were collected and frozen for subsequent ELISA and protein assay. Insulin was detected with

HTRF assay (Cisbio) kits or ELISA (Crystal Chem) and performed on the PHERAstar microplate reader.

## Imaging and data analysis

A commercial inverted Nikon Eclipse Ti microscope equipped with a TIRF objective (Apo TIRF 100×/1.45, oil), ×1.5 magnification, was employed to monitor fusion dynamics[49]. Dispersed pancreatic β-cells are identified by their larger size and unique responsiveness to glucose stimulation[11,51]. The emitted fluorescence collected by Andor iXon DU897 (pixel size, 16 μm). Human pancreatic slices imaging was performed with Leica SP8 lightning confocal. Image processing and quantitative assessments were primarily performed using Fiji software. The data were plotted using IGOR Pro software (v6.11, WaveMetrics) and all figures were organized by Adobe illustrator CS6. For Single-molecule localization and tracking analysis, granule localization was determined by weighted centroid method and tracked with global combinatorial optimization algorithm[17]. Data are presented as mean ± s.d., unless otherwise stated. Statistical significance for all data was determined using two-tailed unpaired Student's *t*-test.

## Reporting summary

Further information on research design is available in the Nature Research Reporting Summary linked to this article.

## Data availability

The authors declare that all data supporting the findings of this study are available within the paper and/or the Supplementary Information. Source data are provided with this paper.

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

## Acknowledgements

This work was supported by a grant from the Canadian Institute of Health Research (CIHR PJT-159741) to H.Y.G. Some of the equipment used in this study was supported by the 3D (Diet, Digestive tract and Disease) Centre funded by the Canadian Foundation for Innovation and Ontario Research Fund, project number 19442. We are most grateful to the Trillium Gift of Life Network, Toronto, Ontario for providing the human pancreases [year 2019-2021], and for the isolated human islets from the Islet Core of the Alberta Diabetes Institute, Edmonton. We thank Sebastian Barg (Uppsala, Sweden) for the Ad-NPY-tdOrange2. The authors thank the Imaging Facility and SPARC Drug Discovery Core Facility, The Hospital for Sick Children, Toronto, Canada, for assistance with imaging and virus packaging. We also thank the Flow Cytometry Facility, Temerty Faculty of Medicine, University of Toronto, Toronto, Canada, for assistance with cell sorting.

## Author contributions

F.K. performed most of the experiments. L.X. conducted electrophysiology studies and physiological analyses. T.Q. assisted with animal husbandry. Y.M. prepared human pancreatic slices and assisted with long-term slice culture. Y.K., T.T., T. L. and H.X. assisted in some of the biochemical studies. All authors discussed the results and contributed to manuscript revision. H.Y.G. and F.K. formulated the original hypothesis and co-wrote the manuscript.

## Competing interests

The authors declare no competing interests.
