## [Peer Review File · Nature Communications]

Plasma membrane flipping of Syntaxin-2 regulates its inhibitory action on insulin granule exocytosisREVIEWER COMMENTS

Reviewer #1 (Remarks to the Author):

This manuscript reported a novel and exciting study exploring the mechanism of stx2 inhibition on insulin secretion. Syntaxin-2 (Stx2) is an inhibitory t-SNARE protein to reduce vesicle fusion intracellularly, and it is also an epimorphin to regulate morphogenesis at the cell surface. The authors proposed an intriguing flip-flop model for stx2 to flip out of the plasma membrane before granule fusion and thus relax its inhibition on insulin release. This model is consistent with the idea that stx2 competes with other syntaxin isoforms intracellularly at vesicle fusion sites to reduce the assembly of fusion-competent SNARE complexes. The authors provided multiple lines of evidence to support the model: 1) Stx2 knockdown (KD) in human islets increased biphasic insulin secretion, granule docking, and newcomer fusion events; 2) 40% of stx2 molecules were colocalized with granules and stx1a; 3) stx2 KD increased the number of trans-SNAREs but reduced cis-SNAREs. 4) The cell surface stx2 increased following glucose stimulation; stx2 flip occurs before exocytosis and is associated with increased fusion probability. 5) Extracellular BoNT/C1-LC treatment increased insulin secretion by cleaving the SNARE motif of flipped stx2 at the cell surface. The reversible stx2-flipping model is innovative as it provides a novel mechanism for stx2 to fine-tune insulin secretion. The manuscript is well written. While a portion of stx2 is well-known to present at the cell surface as an epimorphin, how it reaches the cell surface is unknown. The data presented here suggest that stx2 can flip out to the cell surface and back inside the cells, connecting its two functions at both sides of the plasma membrane. This model may significantly advance the current understanding of stx2-mediated regulation on vesicle fusion. On the other side, this rapid, reversible stx2 flipping is surprising because protein flipping across the cell membrane is rare as it must overcome a significant energy barrier. Several questions need to be addressed to better support this model.

Major points:

- 1) The plasma membrane generally prevents membrane proteins from flipping across the membrane. Although a few proteins can do so, they often have unique topological structures and smaller sizes. Caution is required to test this stx2 flipping model rigorously since stx2 contains a relatively large N-terminal Habc domain and a SNARE module. Explain what makes stx2 unique as compared to the other stx isoforms for its flipping? Does FAP-tag or other fluorescence protein tagged to stx2 affect the flipping? Does stx2 flip across the membrane after dissociating from its clusters or at a cluster form? What percentage of stx2 molecules are present extracellular at resting β cells? How long do they stay on cell surface before flipping back? Stx2 flipping-back requires a better characterization.
- 2) Stx2 KD increased trans-SNARE level and reduced cis-SNARE levels. While these results are consistent with the proposed stx2 function on competing with other syntaxin isoforms for SNARE formation, another possibility is stx2 KD indirectly affecting Munc13 and munc18 activity to change SNARE assembly activity. Do Munc13 and 18 expression levels change in the stx2 KD cells?

Minor points

- 3) The authors used SG "fusion efficiency" in numerous places. Does it mean the SG release probability? How "fusion efficiency" is measured (fig.5c-d)?
- 4) It is important to use non-specific RNAs as negative controls to exclude potential non-specific effect of KD. It is unclear if this is the case in stx2 KD experiments, does it affect insulin secretion?
- 5) In Fig.2c, how is the colocalization analyzed? The figure shows that only a small portion of stx2 was colocalized with granules, and the majority were not, suggesting an excessive stx2 molecules on the membrane. They are well-aligned with the results that stx2 expression level increase did not change its colocalization with granule, but these data are inconsistent with result that stx2 overexpression inhibits insulin release (via competition with syx1a in SNARE assembly).
- 6) Does stx2 KD change other SNARE protein levels? Reduced granule fusion in stx2 KD may also reduce cis-SNARE, which should be considered (fig.3d).
- 7) stx2 flipping events should be characterized in more detail (fig.4). A negative control experiment is required in fig.4c to show the signal changes with time (without glucose stimulation). How fast do they flipped back after stopping glucose stimulation? Is the F/F0 reduction (fig.4d) reflect the stx2 cluster dispersion at the cell surface or a flip-in event? Is the FAP-fluorogen binding reversible? Were the FAP spots overlap with SRB spots or not?
- 8) Described the method used to identify β cells in fig.5a and 5e as there are plenty of non- β cells

in human islets? How often did the flipping events occur at the fusion sites? How was stx2 labeled in Fig.5e-g?

Reviewer #3 (Remarks to the Author):

In a previous study, the same group discovered that deletion of syntaxin-2 moderately increases insulin secretion. Others' groups previously showed that syntaxin-2 could be flipped across the plasma membrane and a proteolytic product serves as a signaling molecule. In this manuscript, the authors followed up on these findings and showed that syntaxin-2 flips across the plasma membrane upon glucose stimulation, which diminishes its inhibitory role in insulin secretion. Overall, this is an interesting finding that sheds light on the molecular machinery mediating insulin secretion.

Major points:

1. Other syntaxins such as syntaxin-1 and 4 are also involved in insulin secretion. The authors need to check if they also flip across the plasma membrane. If they do not, the data would be important controls.

2. How does syntaxin-2 inhibit SNARE assembly? In vitro, syntaxin-2 appears to behave similarly as other SNAREs. More biochemical studies are needed.

3. It is difficult to envision how syntaxin-2 could influence SNARE complex disassembly. A thorough discussion is needed.

4. The authors need to map the domains of syntaxin-2 to determine sequences required for flipping. For example, is the Habc domain or SNARE motif required for flipping?

Minor points:

1. In figures 6 and S5, the expression levels of syntaxin-2 WT and mutants should be shown. The authors did not specify whether endogenous proteins were fully depleted and how the rescue cell lines were made.

Reviewer 1

This manuscript reported a novel and exciting study exploring the mechanism of stx2 inhibition on insulin secretion. Syntaxin-2 (Stx2) is an inhibitory t-SNARE protein to reduce vesicle fusion intracellularly, and it is also an epimorphin to regulate morphogenesis at the cell surface. The authors proposed an intriguing flip-flop model for stx2 to flip out of the plasma membrane before granule fusion and thus relax its inhibition on insulin release. This model is consistent with the idea that stx2 competes with other syntaxin isoforms intracellularly at vesicle fusion sites to reduce the assembly of fusion-competent SNARE complexes. The authors provided multiple lines of evidence to support the model: 1) Stx2 knockdown (KD) in human islets increased biphasic insulin secretion, granule docking, and newcomer fusion events; 2) 40% of stx2 molecules were colocalized with granules and stx1a; 3) stx2 KD increased the number of trans-SNAREs but reduced cis-SNAREs. 4) The cell surface stx2 increased following glucose stimulation; stx2 flip occurs before exocytosis and is associated with increased fusion probability. 5) Extracellular BoNT/C1-LC treatment increased insulin secretion by cleaving the SNARE motif of flipped stx2 at the cell surface. The reversible stx2-flipping model is innovative as it provides a novel mechanism for stx2 to fine-tune insulin secretion. The manuscript is well written. While a portion of stx2 is well-known to present at the cell surface as an epimorphin, how it reaches the cell surface is unknown. The data presented here suggest that stx2 can flip out to the cell surface and back inside the cells, connecting its two functions at both sides of the plasma membrane. This model may significantly advance the current understanding of stx2-mediated regulation on vesicle fusion. On the other side, this rapid, reversible stx2 flipping is surprising because protein flipping across the cell membrane is rare as it must overcome a significant energy barrier. Several questions need to be addressed to better support this model.

Major points:

1) The plasma membrane generally prevents membrane proteins from flipping across the membrane. Although a few proteins can do so, they often have unique topological structures and smaller sizes. Caution is required to test this stx2 flipping model rigorously since stx2 contains a relatively large N-terminal Habc domain and a SNARE module. Explain what makes stx2 unique as compared to the other stx isoforms for its flipping? Does FAP-tag or other fluorescence protein tagged to stx2 affect the flipping? Does stx2 flip across the membrane after dissociating from its clusters or at a cluster form? What percentage of stx2 molecules are present extracellular at resting β cells? How long do they stay on cell surface before flipping back? Stx2 flipping-back requires a better characterization.

We would like to thank the reviewer for the constructive comments. We agree that protein topology change has to overcome the high energy barrier and dynamic topology of an integral membrane protein is challenging the current tacit dogma concerning stable topology once formed.

Real-time imaging of FAP-Stx2 by TIRFM demonstrated that glucose-induced stx2 flipping occurs in less than 1min, and reaches stable maximum after about 15-20 min (see Figure 4 and new Figure S4a-c). The flipped single-clusters indicated by the localized transient fluorescence bursts generally last for seconds to tens of seconds and then dissipate on the cell surface (Figure 4d). In the meantime, the global intensity of cells increases gradually over time (Figure S4c). There is about 15% (derived from $(F_0 - F_{bg}) / (F_{max} - F_{bg})$, F_{bg} indicates the background intensity) of the total Stx2 residing on the extracellular surface. Consistently, population analysis of FAP- accessibility using flow cytometry also showed a similar result, with ~15% being present on the extracellular surface at resting state and the remaining 85% gradually getting flipped over 15-20 min after glucose stimulation (Figure 4e-f), most of which would flip back within 30 min in absence of glucose stimulation if not cleaved (Figure S4d). The 26 kDa FAP tag, a little smaller size than the widely used EGFP, has a little negative effect on the flipping when comparing to overexpressed WT and FAP tagged Stx2. Nevertheless, in the assay used in this study, all constructs included the FAP tag to control and negate any net contribution that could have been induced by the tagging. We updated these descriptions in the manuscript (see Pages 12-14, highlighted in yellow).

Finally, as to why stx2's uniqueness compared to other Stxs for its flipping property, we refer to our detailed responses to Comments 1 and 4 of Reviewer 2.

2) Stx2 KD increased trans-SNARE level and reduced cis-SNARE levels. While these results are consistent with the proposed stx2 function on competing with other syntaxin isoforms for SNARE formation, another possibility is stx2 KD indirectly affecting Munc13 and

munc18 activity to change SNARE assembly activity. Do Munc13 and 18 expression levels change in the stx2 KD cells?

We appreciate the reviewer's comments and have carried out additional experiments as suggested. Munc13-catalyzed conformational changes have been shown to be essential for the transition of the Munc18/Syntaxin complex to the full primed fusion-ready SNARE complex^{1,2,3,4}. Western blots showed that Stx2 knockdown does not affect the expression of Munc13 and Munc18 (new Figure S7d), indicating that the catalytic (priming) chaperones for SNARE complex assembly remain unchanged. In contrast, the native-PAGE study showed Stx2 inhibits the transition of early assembly intermediates to fusogenic SNARE complexes by substituting for fusogenic syntaxins (Stx1 and Stx3, in new Figure 3b). This is confirmed by Stx2 pulling down not only cognate SNAREs but also the priming proteins Munc18 and Munc13 (new Figure S3b); these results are as suspected by this reviewer. This is added to *Results* Section Pages 10-11 highlighted in yellow. Altogether, these results suggest that the two ways Stx2 functioning as i-SNARE is by directly binding to a subunit of fusogenic SNAREpins to form non-fusogenic i-SNAREs or blocking the transition of early assembly intermediates to fusogenic SNARE complex.

Minor points

3) The authors used SG "fusion efficiency" in numerous places. Does it mean the SG release probability? How "fusion efficiency" is measured (fig.5c-d)?

We thank the reviewer for this suggestion. We apologize for the confusion we have caused. To avoid confusion, we would use the term "probability" and corrected the manuscript as well as figures.

The fusion probability in Figure 5c-d calculated by

fusion probability

$$= \frac{\text{predocked and newcomer granules undergoing fusion at sites where flipping occurring}}{\text{total predocked and newcomer granule at sites where flipping occurring}}$$

The bottom axis "..." denote delay time longer than 5 s. We have updated the figure and figure legend to make it clear (Page 38-39, highlighted in yellow).

4) It is important to use non-specific RNAs as negative controls to exclude potential non-specific effect of KD. It is unclear if this is the case in stx2 KD experiments, does it affect insulin secretion?

Thank you for pointing this technical detail out. We apologize for lack of clarity on this issue. Actually, we included both negative and untreated control for shRNA-mediated Stx2 knockdown. The negative control was designed to have no known target to rule out

non-specific effects in RNAi experiment, while the untreated control was designed using the blank vector that both negative control and shRNA-Stx2 cloned into to determine the level of cell viability. Both controls were included when we designing and validating shRNA, while non-targeting negative control was always included in all assay. We have clarified in the *Materials and methods* section (Pages 22-23, highlighted in yellow) and included these controls in supplementary figures (Figure S7d, Page 10 of the supplementary information).

5) In Fig.2c, how is the colocalization analyzed? The figure shows that only a small portion of stx2 was colocalized with granules, and the majority were not, suggesting an excessive stx2 molecules on the membrane. They are well-aligned with the results that stx2 expression level increase did not change its colocalization with granule, but these data are inconsistent with result that stx2 overexpression inhibits insulin release (via competition with syx1a in SNARE assembly).

We thank the reviewer for bringing this up. To emphasize the role of Stx2 clusters in granule fusion, the colocalization was determined by the fraction of docked granules that colocalizes with Stx2 cluster (fluorescent intensity of F/Fs > 20%, surrounding annulus, S.), since there are excessive Stx2 clusters on the membrane. As suggested by the reviewer, to be accurate, the colocalization of granules with Stx2 clusters should be calculated by the docked granules rather all. Thus, we re-examined the colocalization and updated the Figure 2d (see Page 33). Actually, there is no contradiction here, since the colocalized cluster number of Stx2 remained constant whereas the individual clusters became saturated while the expression level of Stx2 was increasing.

6) Does stx2 KD change other SNARE protein levels? Reduced granule fusion in stx2 KD may also reduce cis-SNARE, which should be considered (fig.3d).

We thank the reviewer for bringing up this. At the very beginning, we checked other exocytotic syntaxins, SNAP25, VAMPs and SM proteins by western blotting, which showed no significant altered by Stx2 knockdown. We now included these results in supplementary Figure S7d. The co-IP study and native-PAGE assay (new Figure 3b) showed that Stx2 knockdown enables more *trans*-SNARE complex assembly and consequently more *cis*-SNARE complex formation after fusion, since membrane fusion converts the *trans*-SNARE complex into a *cis*-SNARE complex. The following disassembly assay of ternary SNARE complexes in the presence or absence of Stx2 (Figure 3c) suggested that Stx2 modulates *cis*-SNARE complexes disassembly.

7) stx2 flipping events should be characterized in more detail (fig.4). A negative control experiment is required in fig.4c to show the signal changes with time (without glucose stimulation). How fast do they flipped back after stopping glucose stimulation? Is the F/F0 reduction (fig.4d) reflect the stx2 cluster dispersion at the cell surface or a flip-in event?

Is the FAP-fluorogen binding reversible? Were the FAP spots overlap with SRB spots or not?

We apologize for lack of clarity on this issue. We now included the negative control and details in new Figure S4 (see Page 7 of the supplementary information).

Figure S4 Stx2 flipping characterization and colocalization analysis with exocytosis.

(a) INS-832/13 cell expressing FAP-Stx2 was stimulated with Krebs-Ringer HEPES (KRH) buffer without glucose as negative control for comparison to the glucose-induced flipping shown in Figure 4b. (b) Time lapse images show that no discernible changes of the global FAP-Stx2 observed in the cell in (a). Scale bar, 5 μ m. (c) The time course of FAP-Stx2 flipping in the presence and absence of glucose demonstrated that glucose-induced Stx2 flipping occurs in less than 1 min, and reaches stable maximum after about 15-20 min. (d) Immunoblot evaluating extracellular Stx2 from concentrated supernatants following exposure to 5 μ g/mL BoNT/C1-LC. INS-832/13 cells expressing FAP-Stx2 were incubated with glucose for 15 min and then washed with KRH buffer. (e) Stx2 is able to flip across the plasma membrane and be released into the extracellular space, which is not the case

with Stx1a and Stx4. The indicated 'total' proteins were from total lysate. The indicated 10x concentrated 'extra'-cellular proteins that were released by collagenase treatment of intact islets. (f) Colocalization analysis of exocytosis visualized by SRB with flipping events indicated by FAP-Stx2 in human pancreatic tissue slices comparing to negative control (NC) using a simulated complete spatially random distribution. Maximum intensity projection of SRB events overlapped with FAP-Stx2 hotspots were calculated. Box: 25th and 75th percentiles. Line: median. Whiskers: smallest to the largest values. Markers: average of individual cell.

8) Described the method used to identify β cells in fig.5a and 5e as there are plenty of non- β cells in human islets? How often did the flipping events occur at the fusion sites? How was stx2 labeled in Fig.5e-g?

Up to 50–60 % of the cells in human islets of Langerhans are β -cells⁵, which are responsive to high glucose, compared to less abundant low-glucose responsive α -cells (~30%) and the least abundant δ -cells (~10%). The method used to identify β -cells from dispersed single-cells is based on their larger size and unique response to glucose stimulation. Pancreatic β -cells generally have a larger size (i.e., cell capacitance) which has been shown by its electrophysiological properties, and characteristic electrical and granule-releasing activity in responsiveness to glucose stimulation^{3, 6, 7}. We added some method description accordingly (*Materials and methods* section, Page 29, highlighted in yellow).

Generally, flipping events occurs more frequently and distributes more widely than fusion events. Most (~80 %) fusion events occurred at sites where flipping occurring, while less than 25% of flipping events occurred at fusion sites. We also added some description in main text (Page 14, highlighted in yellow).

The Stx2 labelled in Fig5e-g is Stx2-mScarlet. The legend has been corrected (Page 39, highlighted in yellow). Thank you for pointing this out.

Reviewer 2

In a previous study, the same group discovered that deletion of syntaxin-2 moderately increases insulin secretion. Others' groups previously showed that syntaxin-2 could be flipped across the plasma membrane and a proteolytic product serves as a signaling molecule. In this manuscript, the authors followed up on these findings and showed that syntaxin-2 flips across the plasma membrane upon glucose stimulation, which diminishes its inhibitory role in insulin secretion. Overall, this is an interesting finding that sheds light on the molecular machinery mediating insulin secretion.

Major points:

1. Other syntaxins such as syntaxin-1 and 4 are also involved in insulin secretion. The authors need to check if they also flip across the plasma membrane. If they do not, the data would be important controls.

We thank the reviewer for this comment. As requested by the reviewer, we have performed western blot to evaluate if Stx1a and Stx4 present at extracellular space as a control. This was performed by collagenase releasing of the flipped syntaxins on the extracellular surface, which showed only Stx2 but not Stx1a or Stx4 ended up on the extracellular space. We updated this result in Figure S4e (see Page 7 of the supplementary information), and this mentioned in Page 20 highlighted in yellow.

2. How does syntaxin-2 inhibit SNARE assembly? In vitro, syntaxin-2 appears to behave similarly as other SNAREs. More biochemical studies are needed.

We thank the reviewer for this comment. We performed more biochemical analysis, which demonstrated two possible ways how Stx2 exerts an inhibitory role in insulin secretion. First, our previous co-IP and pull-down study in mouse³ and current single-molecule imaging as well as co-IP in human (new Figure S3b) in this study showed that excessive Stx2 effectively competes with and directly binds to a subunit of SNAREpins to form non-fusogenic SNARE complexes. Second, the native-PAGE assay showed that Stx2 substitutes for fusogenic syntaxins Stx1a and Stx3 to prevent the transition of early assembly intermediates to SNAREpins (new Figure 3b, see Page 35), and stated in *Results* Section in Pages 10-11 highlighted in yellow.

3. It is difficult to envision how syntaxin-2 could influence SNARE complex disassembly. A thorough discussion is needed.

We expanded the discussion and interpretation on how Stx2 could affect *cis*-SNARE disassembly in the revised manuscript (see Page 20, highlighted in yellow).

“*Cis*-SNARE formed by parallel posited α -helices derived from the cognate v- and t-SNAREs within the same membrane after fusion is disassembled through a combined effect of α -SNAP and the ATPase NSF, which allow the SNARE proteins to be recycled⁴. The N-terminal three-helix-bundle of Stx1a has been shown involved in interaction with accessory proteins, such as SM proteins, synaptotagmins as well as complexins, during the multistep SNARE cycling in membrane fusion^{4, 8}. Our single-molecule imaging demonstrated that Stx2 competes with Stx1a to bind docking sites at a higher binding affinity, combining the fact that the sequence of Stx2 shares 70% identity with Stx1a within N-terminal domain, we interpret our results to suggest that the competitive binding of Stx2 to α -SNAP makes the latter binding to *cis*-SNARE less effective, thus reducing the ability of ATPase NSF to effect *cis*-SNARE complex disassembly.”

4. The authors need to map the domains of syntaxin-2 to determine sequences required for flipping. For example, is the Habc domain or SNARE motif required for flipping?

We thank the reviewer for the suggestion to map the domains of Stx2 that required for flipping, although this has been discussed in an earlier study for epimorphin⁹. In that study, the SNARE and TM domain of Stx2 are required for flipping and the conserved key amino acid residue for soluble epimorphin releasing resides in H246⁹. Our flow cytometry assay emphasized the importance of TM domain by showing that the charge alteration of the extra-membrane domain affects the flipping efficiency. Furthermore, we showed targeting the flipping efficiency of Stx2 profoundly modulating insulin secretion, which would benefit restoring the impaired insulin secretion in diabetes. This point is now included in the *Discussion* on Page 20.

Minor points:

1. In figures 6 and S5, the expression levels of syntaxin-2 WT and mutants should be shown. The authors did not specify whether endogenous proteins were fully depleted and how the rescue cell lines were made.

We thank the reviewer for pointing this out. As requested, we have added new Figure S7c, that shows the expression levels of endogenous syntaxin-2 and its mutants by western blot.

We have now improved the *methods* section to clarify how the cell lines were made (see Pages 25-26, highlighted in yellow).

“In order to minimize variation associated with repeated transient transfection and increase reproducibility, all gene-transduced INS-1 cell used in this study, unless specified otherwise, were stable cell lines generated by Lentivirus antibiotic selection. Specifically, 1 mL lentivirus supernatant infected INS-1 cell in 100 mm cell culture dish when its

confluence reaches ~70% in the presence of 10ug/mL of polybrene (Millipore Sigma). The medium was then discarded, and 10 mL of fresh medium added, incubating the cells for 6 hours at 37 °C. Fresh medium containing 2 µg/ml puromycin was applied to select cells expressing the protein of interest 2 days after infection. Single clones were picked after a 4-day 2 µg/ml puromycin selection and then maintained in medium with 1 µg/ml puromycin. Clones were validated by western blot.”

References

1. Wang X, *et al.* Munc13 activates the Munc18-1/syntaxin-1 complex and enables Munc18-1 to prime SNARE assembly. *The EMBO journal* **39**, e103631 (2020).
2. Shu T, Jin H, Rothman James E, Zhang Y. Munc13-1 MUN domain and Munc18-1 cooperatively chaperone SNARE assembly through a tetrameric complex. *Proceedings of the National Academy of Sciences* **117**, 1036-1041 (2020).
3. Zhu D, *et al.* Syntaxin 2 Acts as Inhibitory SNARE for Insulin Granule Exocytosis. *Diabetes* **66**, 948-959 (2017).
4. Jahn R, Scheller RH. SNAREs — engines for membrane fusion. *Nature Reviews Molecular Cell Biology* **7**, 631-643 (2006).
5. Cabrera O, Berman DM, Kenyon NS, Ricordi C, Berggren P-O, Caicedo A. The unique cytoarchitecture of human pancreatic islets has implications for islet cell function. *Proceedings of the National Academy of Sciences* **103**, 2334-2339 (2006).
6. Henquin JC, Meissner HP. Significance of ionic fluxes and changes in membrane potential for stimulus-secretion coupling in pancreatic B-cells. *Experientia* **40**, 1043-1052 (1984).
7. Göpel S, Kanno T, Barg S, Galvanovskis J, Rorsman P. Voltage-gated and resting membrane currents recorded from B-cells in intact mouse pancreatic islets. *The Journal of physiology* **521 Pt 3**, 717-728 (1999).

8. Baker RW, Hughson FM. Chaperoning SNARE assembly and disassembly. *Nat Rev Mol Cell Biol* **17**, 465-479 (2016).
9. Hirai Y, *et al.* Non-classical export of epimorphin and its adhesion to α v-integrin in regulation of epithelial morphogenesis. *J Cell Sci* **120**, 2032-2043 (2007).

REVIEWER COMMENTS

Reviewer #1 (Remarks to the Author):

The revision has addressed all my concerns. I have two minor questions related to the new data presented in revision. 1) Fig.S4 e demonstrated that the flipping is specific to stx2 but not stx1a or stx4. Yet, the extra-stx2 band is much more intense than one would expect. Before collagenase treatment, is the cell culture medium changed? Otherwise, this band will primarily reflect an accumulation of soluble epimorphin rather than the portion of flipped surface stx2 acutely released by collagenase. This result needs clarification with experiment details. 2) in Fig. S2 a-c indicate the clusters of sytx2. Are these clusters present for intracellular stx2, the flipped stx2, or both? Does stx2 flipping occur as the stx2 monomer or in a cluster form?

Reviewer #3 (Remarks to the Author):

The authors addressed my suggestions either with new experiments or additional explanations. In particular, they showed syntaxin1 and syntaxin4 behave differently from syntaxin2. These are important control data that strengthen the conclusions they made. I consider the revision satisfactory.

REVIEWER COMMENTS

Reviewer #1 (Remarks to the Author):

The revision has addressed all my concerns. I have two minor questions related to the new data presented in revision. 1) Fig.S4 e demonstrated that the flipping is specific to stx2 but not stx1a or stx4. Yet, the extra-stx2 band is much more intense than one would expect. Before collagenase treatment, is the cell culture medium changed? Otherwise, this band will primarily reflect an accumulation of soluble epimorphin rather than the portion of flipped surface stx2 acutely released by collagenase. This result needs clarification with experiment details. 2) in Fig. S2 a-c indicate the clusters of sytx2. Are these clusters present for intracellular stx2, the flipped stx2, or both? Does stx2 flipping occur as the stx2 monomer or in a cluster form?

We thank the reviewer for his/her kind assessment and the very constructive comments on our manuscript.

In order to increase sensitivity and highlight the presence of extracellular Stxs, we collected the accumulated extracellular supernatant during the 30-min glucose stimulation and concentrated 10-fold after collagenase treatment. We have now improved the details describing the sample collection in figure legends (Page 7 of the supplementary information, highlighted in yellow).

Although, technically speaking, there is indeed some possibility to label both intracellular and flipped Stx2, we believe that the clusters in Figure S2 a-c indicated by Stx2-mScarlet are intracellular Stx2, based on the following facts: (1) Low possibility of flipping at resting state. The cells in Fig S2 a-c we used for bleaching were maintained in resting state before fixation with 4% PFA. As the response we made to your previous comments, flow cytometry showed that most Stx2 remain at the intracellular surface at resting state; (2) Duration of the fluorescent puncta. Comparing the FAP-accessibility imaging (Figure 4) with live-cell single-molecule imaging (Figure 2 a-c), the mScarlet-tagged Stx2 puncta remained stable for several minutes with somewhat free lateral movement, behaving similar to Stx1a puncta, while the flipped Stx2 indicated by the FAP-fluorogen localized transient fluorescence bursts generally last for seconds to tens of seconds and then dissipate.

According to the intensity of FAP-fluorogen puncta in Figure 4b-d and the overlap of the localized transient bursts with the area of global fluorescent increase when performing z-projection to average the FAP-fluorogen signal, we believe that the majority of the Stx2 flips in a cluster form, although we cannot rule out the possibility of some monomer Stx2 flipping. We thank the reviewer for the incisive comments and hope our response the reviewer finds satisfactory.

Reviewer #3 (Remarks to the Author):

The authors addressed my suggestions either with new experiments or additional explanations. In particular, they showed syntaxin1 and syntaxin4 behave differently from syntaxin2. These are important control data that strengthen the conclusions they made. I consider the revision satisfactory.

Thank you for your approval. We appreciate all the very insightful advice from the reviewer, which greatly improved the manuscript.

References

1. Wang X, *et al.* Munc13 activates the Munc18-1/syntaxin-1 complex and enables Munc18-1 to prime SNARE assembly. *The EMBO journal* **39**, e103631 (2020).
2. Shu T, Jin H, Rothman James E, Zhang Y. Munc13-1 MUN domain and Munc18-1 cooperatively chaperone SNARE assembly through a tetrameric complex. *Proceedings of the National Academy of Sciences* **117**, 1036-1041 (2020).
3. Zhu D, *et al.* Syntaxin 2 Acts as Inhibitory SNARE for Insulin Granule Exocytosis. *Diabetes* **66**, 948-959 (2017).
4. Jahn R, Scheller RH. SNAREs — engines for membrane fusion. *Nature Reviews Molecular Cell Biology* **7**, 631-643 (2006).
5. Cabrera O, Berman DM, Kenyon NS, Ricordi C, Berggren P-O, Caicedo A. The unique cytoarchitecture of human pancreatic islets has implications for islet cell function. *Proceedings of the National Academy of Sciences* **103**, 2334-2339 (2006).
6. Henquin JC, Meissner HP. Significance of ionic fluxes and changes in membrane potential for stimulus-secretion coupling in pancreatic B-cells. *Experientia* **40**, 1043-1052 (1984).

7. Göpel S, Kanno T, Barg S, Galvanovskis J, Rorsman P. Voltage-gated and resting membrane currents recorded from B-cells in intact mouse pancreatic islets. *The Journal of physiology* **521 Pt 3**, 717-728 (1999).
8. Baker RW, Hughson FM. Chaperoning SNARE assembly and disassembly. *Nat Rev Mol Cell Biol* **17**, 465-479 (2016).
9. Hirai Y, *et al.* Non-classical export of epimorphin and its adhesion to α v-integrin in regulation of epithelial morphogenesis. *J Cell Sci* **120**, 2032-2043 (2007).

REVIEWERS' COMMENTS

Reviewer #1 (Remarks to the Author):

The authors have addressed all my questions.